# On the Connection between Pre-training Data Diversity and Fine-tuning Robustness

**Vivek Ramanujan**[*][†]    **Thao Nguyen**[*][†]
**Sewoong Oh**[†]    **Ludwig Schmidt**[†◇]    **Ali Farhadi**[†◇]
[†]University of Washington    [◇]Allen Institute for AI
{thaottn,ramanv}@cs.washington.edu

## Abstract

Pre-training has been widely adopted in deep learning to improve model performance, especially when the training data for a target task is limited. In our work, we seek to understand the implications of this training strategy on the generalization properties of downstream models. More specifically, we ask the following question: how do properties of the pre-training distribution affect the robustness of a fine-tuned model? The properties we explore include the label space, label semantics, image diversity, data domains, and data quantity of the pre-training distribution. We find that the primary factor influencing downstream effective robustness [44] is data quantity, while other factors have limited significance. For example, reducing the number of ImageNet pre-training classes by $4\times$ while increasing the number of images per class by $4\times$ (that is, keeping total data quantity fixed) does not impact the robustness of fine-tuned models. We demonstrate our findings on pre-training distributions drawn from various natural and synthetic data sources, primarily using the iWildCam-WILDS distribution shift as a test for robustness.

## 1 Introduction

Transfer learning is a popular technique to deal with data scarcity, improve training speed, or transfer useful inductive biases that can benefit downstream tasks [28, 8, 10]. In the domain of computer vision, pre-training on ImageNet in particular has been the de-facto standard for obtaining features to solve a wide range of vision tasks, such as object detection [34, 9, 18], segmentation [5, 16], and action recognition [42]. While there exists previous work that seeks to pinpoint specific properties of ImageNet-trained features that benefit downstream performance [19, 24, 23, 38], the analysis is often done with respect to model accuracy. Our work instead examines the robustness of fine-tuned models to natural distribution shifts. Instead of looking at architecture variations and pre-training algorithm as done in prior work [38, 14, 49], we focus on the role of the pre-training data. This data-centric approach has been validated by past work [27, 29, 13], which show that the training data distribution plays a larger role than training methods or architecture in influencing model robustness.

Robustness under distribution shifts is a fundamental concern for producing reliable machine learning systems: a model can perform in unexpected and undesirable ways when there is a mismatch between the data distribution encountered in deployment and the one on which the model is trained [36, 22]. For example, a self-driving car should be able to generalize to a wide variety of weather scenarios to be considered safe, some of which it may not have seen during training. In our work, we focus on these forms of *natural* distribution shifts [33, 22]—named so because they are induced by real-world processes—and study what aspects of the source dataset could help fine-tuned models become more robust to these shifts. We tackle this question along five different ablation axes: **(i)** Data quantity, **(ii)** Label granularity, **(iii)** Label semantics, **(iv)** Image diversity, and **(v)** Data sources. Through a

---

[*]Equal contribution

37th Conference on Neural Information Processing Systems (NeurIPS 2023).

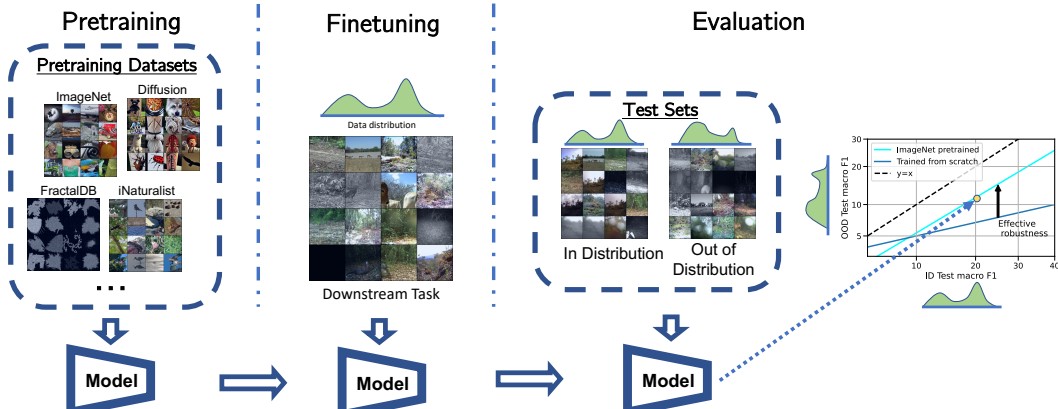

Figure 1: A summary of our experimental pipeline. We pre-train a model on a variety of different data distributions and evaluate its effective robustness after fine-tuning on a downstream task (i.e., iWildCam). By examining many models in this manner, we can determine empirical properties of the pre-training distribution that are important for fine-tuning robustness.

better understanding of the interplay between various properties of the pre-training distribution and downstream robustness, we seek to establish guidelines for constructing better pre-training datasets for fine-tuning.

Previous work by Miller et al. [27] experimented with a wide range of natural distribution shifts and found that pre-training on ImageNet yields the biggest improvement in robustness for the iWildCam-WILDS dataset [22, 2]. Consequently, we use iWildCam-WILDS as a probe to evaluate how our interventions with the ImageNet pre-training distribution would alter the robustness trend uncovered in this previous work. We also analyze the use of other pre-training data sources that may differ significantly from ImageNet in both semantic content and data collection methodology. Our main findings can be summarized as follows:

(i) **Data quantity.** Pre-training with more data helps boost robustness. However, we do not need a lot of pre-training data to see significant robustness gains: using 25K images subsampled from either ImageNet or iNaturalist, which is 6× smaller than the size of the fine-tuning dataset, already offers noticable robustness improvements.

(ii) **Label granularity.** Making labels more coarse-grained lowers transfer robustness. The effect is less significant than altering data quantity: extreme reduction in label granularity (e.g., using 5 coarse classes instead of 1000 fine-grained classes) still preserves some of the robustness gains compared to training from scratch.

(iii) **Label semantics.** Given enough data and labels, using more semantically similar classes does not have a notable impact on the robustness of fine-tuned models. In particular, we find that pre-training on the 600 inanimate object categories in ImageNet yields the same effective robustness as pre-training on the 400 animal categories, despite the fact that the downstream task consists of only animal categories.

(iv) **Image diversity.** Given the same pre-training label set and data quantity, increasing per-class diversity (e.g., by including more subclasses) has no effect on transfer robustness. In addition, the trade-off between having more classes and more images per class is not significant if total number of samples is kept constant.

(v) **Data sources.** We find that natural data sources (i.e., ImageNet, iNaturalist) yield similar downstream robustness when controlling for data quantity. Pre-training with synthetic fractal data is less effective at the same data quantity regime but still has some robustness gain to offer compared to training from scratch. Synthetic natural-looking data (e.g., generated by Stable Diffusion [35]) can help close this gap between using natural data and synthetic fractal data.

Overall we find that increasing pre-training data quantity and label granularity makes fine-tuned models more robust to distribution shifts. However, not all additional data is equally helpful. For instance, in the context of iWildCam-WILDS task, pre-training with natural-looking data offers much more robustness than using 10× more synthetic fractal data.

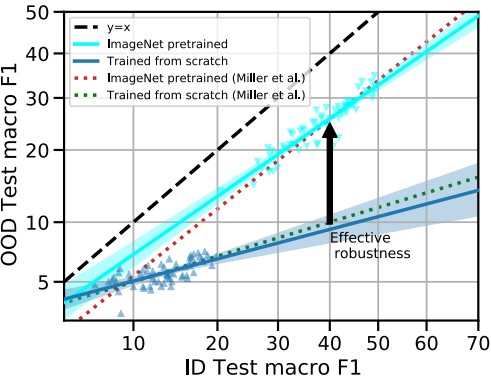

Figure 2: Effective robustness is defined as movement towards a classifier which is robust to distribution shift (i.e., line $y = x$). Using this metric, Miller et al. [27] observes that for the iWildCam-WILDS task, models pre-trained on ImageNet are much more robust than models trained from scratch. We reproduce these two trends and use them as points of reference for our subsequent experiments, in which we modify the pre-training distribution and observe how our interventions alter the robustness trend lines.

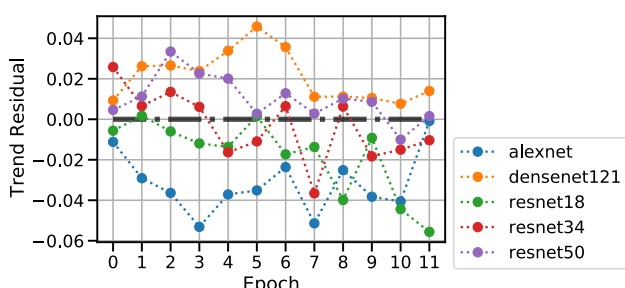

Figure 3: We visualize the residuals of various architectures after fitting a linear trend that predicts OOD accuracy from ID accuracy. All models are pre-trained on the full ImageNet dataset and fine-tuned on iWildCam for 12 epochs. We observe that overall the residuals fluctuate around the $y = 0$ line and vary throughout the course of fine-tuning for most architectures.

## 2 Background

The main motivation for our paper comes from the work by Huh et al. [19], which investigates various factors in ImageNet training that affect the quality of the features used subsequently for transfer learning. For our investigation, we shift the focus from accuracy to robustness against distribution shift, which has been a long-standing issue in machine learning [31, 43, 3, 4]. In particular, we analyze the robustness of pre-trained features to natural distribution shifts observed in the real world through the iWildCam-WILDS benchmark [22]. Furthermore, in contrast to Huh et al. [19], we experiment with a greater variety of more recent neural network architectures, in addition to exploring the use of synthetic pre-training data.

A key goal in robustness is to reduce the impact of distribution shifts on the performance of a model. If model performances on in- and out-of-distribution test sets are plotted along the $x$ and $y$-axes of a scatter plot respectively, then a more robust model would lie closer to the diagonal $y = x$ line. This notion of robustness was captured by Taori et al. [44] under the term *effective robustness*, which measures the difference between a model's actual OOD performance and what could be predicted from its ID performance (Figure 2). Miller et al. [27] adopted this effective robustness framework and evaluated hundreds of models on various distribution shift settings. The authors observed that ID performance is highly correlated with OOD performance. This linear trend mapping ID to OOD performance, and how close it is to the $y = x$ line, is what we use in our work to compare the quality of the pre-trained features.

More notably, Miller et al. [27] discovered that on the iWildCam dataset, models trained from scratch and those that have been pre-trained on ImageNet lie on distinct linear trends, with the latter exhibiting more robustness. We replicate these reported trends in Figure 2. Motivated by this result, our work seeks to better understand what aspects of ImageNet pre-training contribute to this improved robustness on iWildCam, and how these aspects translate to other pre-training data sources.

Previous work by Andreassen et al. [1] has looked at effective robustness over the course of fine-tuning and found that pre-trained models exhibit high effective robustness in the middle of fine-tuning, which eventually decreases as the training proceeds. The paper also experimented with ImageNet as one of the pre-training data sources. In our investigation, as a sanity check to remove number of training epochs as a potential source of bias for the linear fit, we adopt the linear trend of models pre-trained on ImageNet and fine-tuned on iWildCam computed previously by Miller et al. [27] as the baseline. We then report the residuals from comparing actual OOD performance at different epochs to what could be predicted from the corresponding ID performance using this baseline. Refer

| | Training set size | Number of classes | Class distribution | Class hierarchy | Expert-labeled |
|---|---|---|---|---|---|
| ImageNet | 1,281,167 | 1,000 | Class-balanced | WordNet | No |
| iNaturalist | 579,184 | 5,089 | Long-tailed | Tree of life | Yes |

Table 1: Differences between the ImageNet and iNaturalist datasets.

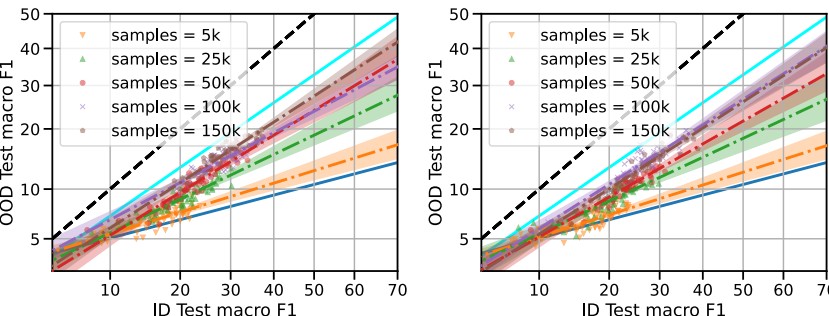

Figure 4: Reducing the number of pre-training images randomly sampled from **(left)** ImageNet and **(right)** iNaturalist lowers the robustness linear trends of the fine-tuned models. However, note that using only 25K pre-training samples (green line) still yields significant robustness improvements compared to training from scratch on 129K iWildCam images (dark blue line). We subsample iNaturalist to ensure class balance, by only including 1000 classes with the most number of samples.

to Figure 3 for more details. We find that in the context of iWildCam fine-tuning, at each epoch, the residuals from our architectures of choice concentrate around the $y = 0$ line and exhibit no particular trend. This in turn allows us to vary the number of fine-tuning epochs as a hyperparameter, and obtain models covering a wide range of test performances for the scatter plots.

## 3 Experimental Setup

As mentioned earlier, the downstream task of interest is wildlife classification with the iWildCam-WILDS dataset [22]: the input is a photo taken by a camera trap, and the output is one of 182 different animal species. There are two test sets for evaluation: ID test data consists of images taken by the same camera traps as the training set, but on different days from the training and validation (ID) images. In contrast, OOD test data contains images taken by a disjoint set of camera traps from training and validation (ID) images. We include some examples of the geodiversity represented in each test split in Appendix Figure 13. Following [22], we report the macro F1 scores of the trained networks because this metric emphasizes performance on rare species, which is critical to the biodiversity monitoring application that the dataset was designed for.

**Pre-training datasets.** We use ImageNet [11] and iNaturalist [46] as the primary pre-training distributions of interest, given their hierarchical structures, complexity, and relevance to the downstream task. The two data sources also differ in many ways (Table 1), hence their pre-trained features make for an informative comparison. We also include experiments with synthetic pre-training data by using Stable Diffusion [35] and the FractalDB-1k dataset [21]. We will elaborate on this in Section 4.5.

**Network architectures.** To obtain data for plotting linear trends, we train a range of standard neural network architectures including ResNet [15], ResNext [48], DenseNet [20], AlexNet [25] and MobileNet V3 [17]. In our scatter plots, besides varying the architectures, we also vary the number of fine-tuning epochs to obtain models with varying F1 scores. Appendix A contains further training details. While our focus is on supervised pre-training, we also report some additional results with CLIP [32] architecture in Section 5.

In the subsequent sections, we detail different interventions made to the pre-training distribution to disentangle key properties of interest. We show the resulting linear trends in relation to the trends replicated from previous work [27], which include models trained from scratch on iWildCam (solid blue line) as well as models pre-trained on ImageNet (solid cyan line). For each trend line, we show 95% bootstrap confidence intervals for the linear fit.

## 4 Experiment Results

### 4.1 Effect of Data Quantity

First, we experiment with reducing the pre-training set size. To remove potential confounding effects from a long-tailed data distribution, we ensure that the class distribution of our pre-training datasets

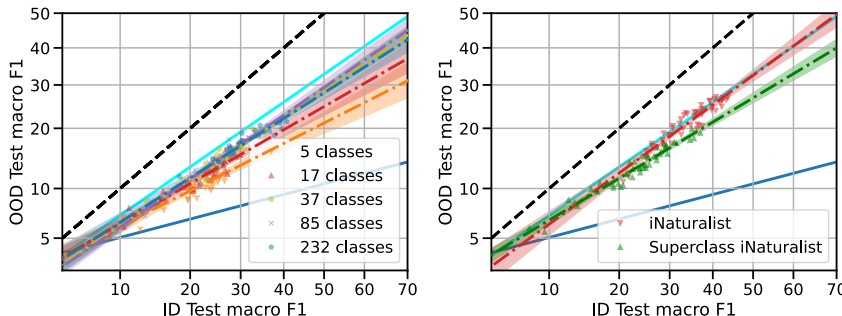

Figure 5: Results of changing the label granularity of the pre-training task by combining classes according to some semantics hierarchy to form supersets, for **(left)** ImageNet and **(right)** iNaturalist. In general, this intervention lowers model robustness on downstream task. However, extreme reduction of the pre-training label space, e.g. by 200× in the case of ImageNet, still offers robustness gains compared to training from scratch.

are uniform. ImageNet is already class-balanced, but this is not the case for iNaturalist [46]. We experiment with a 1000-class subset of iNaturalist using its most frequent classes. We further select images within each class uniformly at random so that the number of samples is the same across all classes. This results in a class-balanced training set of size 150K from iNaturalist. We repeat the same procedure to obtain subsets of size 100K, 50K, 25K and 5K. A similar subsampling process is done on ImageNet, using the full 1000-class dataset which already has a uniform label distribution.

In Figure 4, we observe that reducing the data quantity during pre-training lowers the effective robustness of fine-tuned models. However, it is worth noting at 25K images, pre-training with subsampled ImageNet and iNaturalist data still produces much more robust models compared to training from scratch. This is 6× less data compared to what is used for fine-tuning. As a sanity check, we find that using only 5K samples (i.e., 5 examples per class) during pre-training yields roughly the same level of robustness as training from scratch on iWildCam.

## 4.2 Effect of Label Granularity

Next, we adapt a question raised previously by Huh et al. [19] to our investigation: how does varying the number of pre-training classes affect downstream robustness? Following [19], we construct *supersets* of classes in ImageNet using the WordNet hierarchy. We use the maximum of the shortest path distance from the root of WordNet to a label to compute the maximum depth of the current label set. We then contract ImageNet label nodes along the shortest path to construct superclasses. Specifically, we investigate depths 2, 4, 5, 6, and 7, which result in class counts of 5, 17, 37, 85, and 232 respectively, in order to provide good coverage across a range of label granularities. Similarly, on iNaturalist, we use the superclass information that comes with the dataset to collapse the label space from 5,089 fine-grained classes to 13 coarse classes.

For ImageNet pre-training, we find that using the full 1000 classes provides the most robustness. However, when the label set size is reduced by four times (i.e., taking 232 superclasses at depth 7), model robustness only decreases slightly. From then on, reducing the label set further to 85 classes (depth 6), and then 37 classes (depth 5), does not deteriorate the linear trend further. Only when we experiment with 17 classes (depth 4) do we find another noticeable reduction in effective robustness. With 5 superclasses as the only pre-training labels (depth 2), pre-trained models still yield significantly more robustness than training from scratch.

On iNaturalist, we also observe a similar downward shift in linear trend when we reduce the initial label space to its phylum. Refer to Figure 5 for more details. Overall these findings suggest that using fine-grained labels during pre-training is better for learning representations that are robust to distributions shifts in downstream tasks. But even if only coarse labels are available, pre-training with enough data still has significant robustness benefits to offer.

## 4.3 Effect of Label Semantics

The number of pre-training classes seems to have an impact on downstream robustness, but does it matter what kind of classes models are pre-trained on? We next investigate whether using classes whose semantics are more aligned with the downstream task would improve robustness.

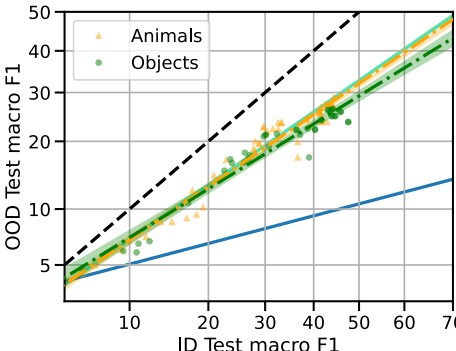

Figure 6: Varying the semantic category of classes included in the pre-training data yields similar robustness linear trends, with pre-training only on "animal" classes exhibiting slightly higher F1 scores than pre-training only on "object" classes. Even though the downstream task is animal classification, models pre-trained only on "object" classes are still much more robust than models that do not undergo any pre-training.

To do so, we separately pre-train models on ImageNet classes that are subsets of the "object" and "animal" WordNet synsets. This yields 2 broad categories that are similar in total sample size, each having around 600K images. In Figure 6, we find that models trained on "animal" classes (yellow line) exhibit slightly higher F1 scores, but roughly the same effective robustness as models trained on "object" classes (green line). This is surprising given that the fine-tuning distribution, iWildCam, contains only images of animals in the wild, which are semantically more similar to the animal classes of ImageNet. It is worth noting that models pre-trained on "object" classes are also much more robust than models trained from scratch (blue line).

One potential confounder to this experiment setup is that some images from "object" classes also contain animals (i.e., co-occurrences that are not accounted for by ImageNet labels). To estimate the extent of this problem, we use TensorFlow's ImageNet2012 multilabel set [41], containing 20k ImageNet validation data with multi-class labels reviewed by experts. We find that 1.1% of the data have labels from both "animal" and "object" classes present in the same image, suggesting that the label co-occurrence issue only impacts a small fraction of training data. Consequently, we posit that training on a diverse set of classes in general helps the model pick up on useful invariances that in turn lead to similar downstream robustness. We explore this hypothesis further in Section 4.5 with synthetic training data.

## 4.4 Effect of Image Diversity

Besides labels, another source of diversity comes from the training images themselves. We experiment with two different notions of image diversity : **(i)** Label diversity, and **(ii)** Per-class image diversity.

### 4.4.1 More Data Per Class vs. More Classes of Data

A natural question arises when designing a dataset with a fixed data budget (or labeling cost): *should we collect more data from existing categories or more categories of data?* To address this question, we keep the total number of images fixed while varying the number of classes of ImageNet we use for pre-training. For example, if we have a budget of 60K images and 100 randomly selected ImageNet classes, we sample 600 images uniformly at random from each of these classes (Figure 7). Here, we find that in the context of the iWildCam distribution shift, there is no difference on downstream robustness between having more data per class or having more classes, as long as the total number of images is constant. This observation also holds at a larger data quantity scale (300K images, see Appendix Figure 18). This result demonstrates the dominant effect of data quantity over other aspects of the pre-training distribution (e.g., label set size).

### 4.4.2 Image Diversity Within Each Class

Another way of modulating dataset diversity is by changing *per-class* image diversity. For example, given a "dog" class, a dataset which only contains images of one dog breed could be seen as less diverse than a dataset which has examples of several breeds. In order to construct a controlled experiment, we use a quantitative heuristic for the diversity of each class: we fix certain superclasses (using the WordNet hierarchy) as the training labels and vary the number of corresponding subclasses where the images are taken from. For iNaturalist we can do the same with the tree of life structure. More diversity means more subclasses chosen per superclass.

For the ImageNet distribution that is built on the WordNet hierarchy, we construct a subset following BREEDS [39]. The two main ways that BREEDS recalibrates the WordNet hierarchy fit our goals for image diversity: (i) selected nodes convey some visual information, and (ii) nodes of similar

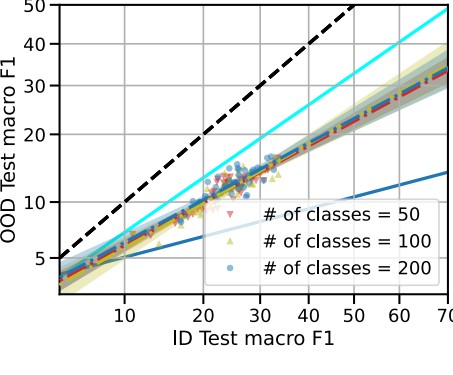

Figure 7: We vary the number of classes randomly selected from the original 1000 ImageNet classes and adjust the number of images per class correspondingly, such that total image quantity is 60K. We observe that having 4× more classes, or 4× more images per class, induces the same level of robustness in fine-tuned models. Experiments with 300K data regime can be found in Appendix Figure 18.

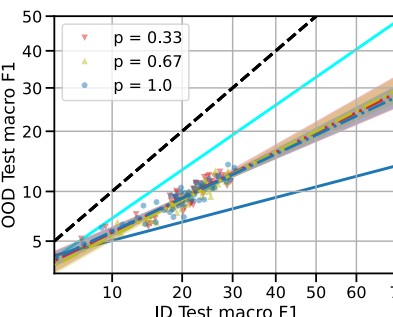
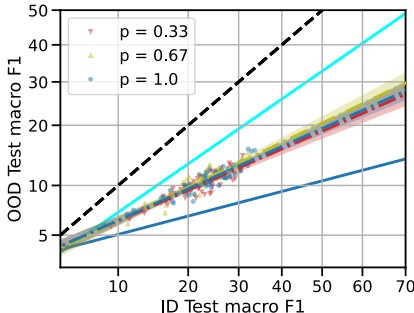

Figure 8: We fix the total amount of pre-training data and the label space, while reducing the number of subclasses that constitute each superclass label in **(left)** ImageNet and **(right)** Naturalist. Smaller $p$ (diversity ratio) means proportionally fewer subclasses per superclass. We find that reducing per-class diversity by up to 3× has no effect on the robustness of downstream models.

specificity share the same distance from the root (e.g., "dog" and "cat" are now both at the same depth even if the "dog" subtree is much larger). With this new hierarchy, we obtain 16 superclasses, each encompassing 12 subclasses (i.e., original ImageNet classes). The full list can be found in Appendix C.1. We vary image diversity by changing the number of subclasses per superclass: 4, 8 or 12 subclasses—corresponding to the diversity ratio $p = 0.33, p = 0.67, p = 1.0$ in the left panel of Figure 8. To prevent data quantity from being a confounding variable, we subsample images from each chosen subclass accordingly (e.g., if we reduce number of subclasses per superclass by 3× then we sample 3× more images from each subclass).

For iNaturalist, we fix the total number of images at 80K and apply the same procedure described above to select a fraction of subclasses (see diversity ratio values in the right panel of Figure 8), for each of the following superclasses: "Plantae", "Insecta", "Mammalia", "Fungi", "Aves", "Reptilia", and "Amphibia." We choose this set of superclasses so we could have a uniform distribution of images per class while maintaining the same number of images as our ImageNet experiment. For more details, see Appendix C.1. As seen in Figure 8, for both ImageNet and iNaturalist, the resulting linear trends are highly similar regardless of the diversity ratio $p$, or the number of subclasses per superclass. We conclude that in this case, per-class image diversity does not have a significant impact on downstream robustness. Note that this does not hold in the extreme setting, e.g. repeating the same image to produce a dataset.

### 4.5 Pre-training with Different Data Sources

Moving beyond interventions *within* each data distribution, we now compare fine-tuning robustness behaviors *across* different data sources.

Compared to ImageNet, iNaturalist exhibits different characteristics on multiple axes (see Table 1). We expect that pre-training on the diverse, domain-specific species – which have been verified by nature enthusiasts – in iNaturalist will provide a boost on robustness for the downstream animal-in-the-wild classification task, compared to training on general web-curated classes in ImageNet. However, in Figure 10, we find that iNaturalist behaves similarly to ImageNet as a pre-training data source. Even when we subsample iNaturalist to follow the ImageNet class distribution (refer to Section 4.1 for its construction), we observe a similar level of effective robustness compared to

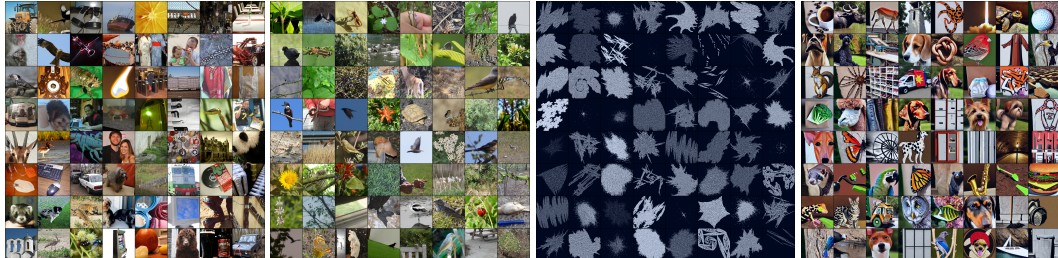

Figure 9: Each grid in order shows random examples from the ImageNet ILSVRC 2012 challenge train set [37, 11], the iNaturalist 2017 challenge train set [46], the FractalDB-1k synthetic train set [21], and a 1000-class ImageNet-style synthetic dataset generated using Stable Diffusion [35].

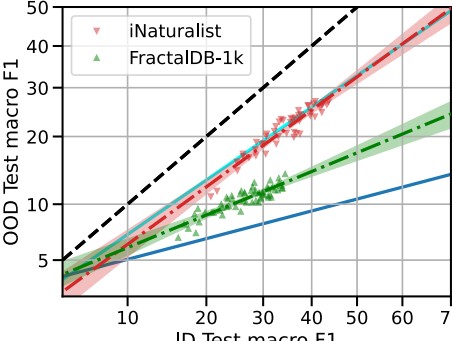

Figure 10: Pre-training on a noisy, long-tailed distribution of natural images like iNaturalist (red line) does not change the robustness on downstream task, compared to pre-training on a clean, class-balanced dataset like ImageNet (cyan line). Pre-training on the same amount of synthetic fractal data (FractalDB-1k) yields much lower robustness (green line), but still has some benefits compared to training from scratch (dark blue line).

the equal-sized 150K ImageNet subset (Figure 11). We hypothesize that when a certain level of "diversity" is reached with the training images and labels, there is negligible robustness gain to be made even if we increase the alignment between the pre-training and fine-tuning data domains.

### 4.5.1 Synthetic Data Sources

To push our diversity hypothesis to the limit, we also pre-train the same set of architectures on FractalDB-1k dataset [21], which has similar class distribution to ImageNet but only contains synthetic fractal images. Pre-training on FractalDB-1k has been shown to surpass the accuracy of pre-training on ImageNet/Places [21]. For the task of iWildCam-WILDS, however, it is noticeably less effective at improving downstream robustness compared to natural image data (Figure 10). However, pre-training with fractal images still offers more robustness than training from scratch.

Can we generate better synthetic data for pre-training than FractalDB-1k? We experiment with Stable Diffusion [35], a popular diffusion model which generates high-quality images from natural language prompts, to generate natural-looking images following the ImageNet class distribution. We use 80 diverse prompts per ImageNet class from [32] to generate a 150K ImageNet-like dataset. Examples from this synthetic dataset can be seen in Figure 9. We find that pre-training on this dataset yields similar robust generalization behaviors as using the same quantity of ImageNet and iNaturalist images (Figure 11). However, at a larger scale of 1M images, the robustness benefits of pre-training with synthetic data begins to saturate and slightly lag behind iNaturalist and ImageNet. See Appendix Figure 19 for more details.

Overall our findings demonstrate that while nuances in image semantics during pre-training are not important for fine-tuning robustness (e.g., Animals versus Objects classes, or iNaturalist versus ImageNet), it is still beneficial to match the general characteristics of the downstream data (e.g., "natural-looking" images).

## 5 Self-supervised Pre-training

Previous experiments revolve around the supervised learning settings. However, it is increasingly common to pre-train on *self-supervised* tasks using web-crawled corpora, which has been shown to significantly improve robustness to distribution shifts [32]. Our preliminary experiments with pre-trained CLIP models [32] on iWildCam show that the resulting ID and OOD performances still lie on the ImageNet pre-training linear trend (i.e., cyan line), despite the self-supervised training mechanism and the much larger training dataset of CLIP. Varying CLIP's data sources only move the F1 scores along the same line (Figure 12).

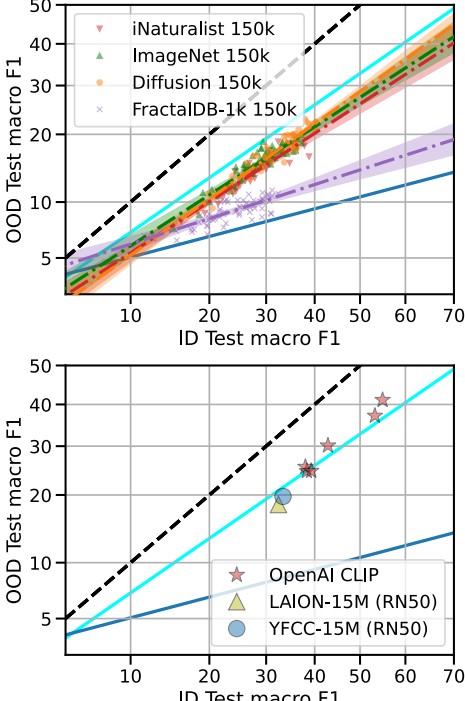

Figure 11: Similar to results in Figure 10, with a budget of 150K images, pre-training on FractalDB-1k is significantly less effective than iNaturalist or ImageNet. However, we find that generating natural-looking data with Stable Diffusion can help close this gap. All datasets are subsampled to be class-balanced, each having 1000 classes.

Figure 12: Results from fine-tuning CLIP [32] pre-trained models on iWildCam-WILDS. The models we use include CLIP with ResNet-50 image encoder trained on YFCC-15M [45] and LAION-15M [40] respectively, as well as all original CLIP models released by OpenAI, including 3 with ViT [12] backbone, trained on a dataset of size 400M. All of these models lie on the same linear trend as that of ImageNet pre-training, demonstrating the consistency of this trend across many pre-training dataset size scales and training algorithms.

We also repeat the experiments with varying data quantity (Section 4.1) in the context of CLIP's image-text pre-training data. Refer to Appendix E for more details. We leave an extensive evaluation of how "diversity" of the pre-training distribution should be defined and measured differently on these open-vocabulary web datasets to future work.

## 6   Conclusion & Discussion

In this work, we find that many factors during pre-training such as label semantics and image diversity, do not significantly alter the effective robustness of models fine-tuned on iWildCam-WILDS. The more influential factors for downstream robust generalization are the *quantity* of the pre-training data and the *granularity* of the label set. Through experiments with Stable Diffusion, we also demonstrate the potential of synthetic natural-looking images as a way to increase the effectiveness of the pre-training distribution along these two ablation axes.

We can think about pre-training dataset construction in terms of an explore vs. exploit trade-off. Exploration, such as finding new data sources, is often time consuming, while exploiting, or collecting as much data as possible from an existing source, can sometimes be significantly easier. Our experiments suggest that a good approach to building pre-training dataset for robust generalization is to find a few data sources that exhibit robustness characteristics on a downstream task (e.g., Stable Diffusion data), and then collect as many data points from these sources as possible.

It is important to note that we are studying a different model behavior than Huh et al. [19]. Some interventions can reduce average performance while maintaining effective robustness (e.g., label granularity in Figure 5) and vice-versa (e.g., architecture modifications). Thus, certain choices during pre-training dataset design depend fundamentally on the goals of the dataset. For example, whether we want to achieve consistent performance across many settings (i.e., robustness) or optimize for very good performance on one specific application changes the applicability of our results.

An important open question is determining what characteristic of the iWildCam-WILDS task leads to the difference in linear trend between pre-training and training from scratch. Some other datasets (e.g., fMoW-WILDS, see [22, 7]) do not exhibit this behavior after fine-tuning, so it is important to pinpoint distribution shifts where pre-training can provide a significant boost in effective robustness. Finding a unifying property among such datasets would allow for better interpretation of our current results. In Appendix G, we look into distribution shift settings constructed from the DomainNet benchmark [30], and observe that pre-training and training from scratch only produce different linear trends for certain pairs of domains and not others. Refer to this Appendix for further discussion.

## 7 Acknowledgements

This work is supported in part by Open Philanthropy, the Allen Institute for AI, and NSF grants DMS-2134012 and CCF-2019844 as a part of NSF Institute for Foundations of Machine Learning (IFML). Ali Farhadi acknowledges funding from the NSF awards IIS 1652052, IIS 17303166, DARPA N66001-19-2-4031, DARPA W911NF-15-1-0543, and gifts from Allen Institute for Artificial Intelligence, Google, and Apple.

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
