# Appendix

## A    Training Details

For the standard deep neural network architectures used in our investigation, their implementation comes from PyTorch's *torchvision.models* package. We use standard hyperparameters found in the WILDS and PyTorch's official GitHub repositories (for training models on iWildCam-WILDS and ImageNet respectively), which match the hyperparameters described in [15] and [22]. Pre-training on different data sources all uses 90 epochs in total, and then models are fine-tuned on iWildCam for 12 epochs.

## B    Dataset Details

**iWildCam-WILDS**    [3] The iWildCam dataset consists of images of 182 animal species, which are captured through the use of camera traps. We use the iWildCam version 2.0 released in 2021 as a correction to the iWildCam 2020 competition dataset [2] to prevent test set leakage. To construct a natural distribution shift, we follow the split proposed by Koh et al. [22], which results in 2 test sets for evaluation: ID test data consists of images taken by the same camera traps as the training set, but on different days from the training and validation (ID) images, while OOD test data contains images taken by a disjoint set of camera traps from training and validation (ID) images. The train set consists of 129,809 images. The distribution of animal categories over these images is long-tailed (see [22]). The ID test set consists of 8,154 images and the OOD test set consists of 42,791 images, also not class balanced. Examples of train set images can be seen in Figure 14.

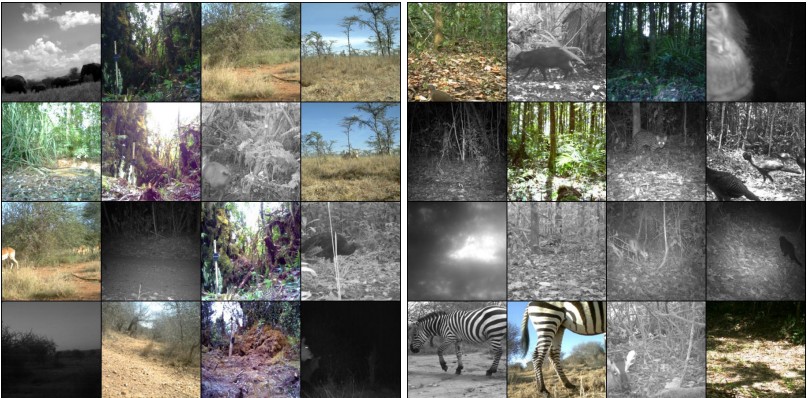

Figure 13: **(left)** Random images from the in-distribution test set of iWildCam-WILDS [22]. **(right)** Random examples from the out-of-distribution test set. This split is based on the geolocations of the camera traps.

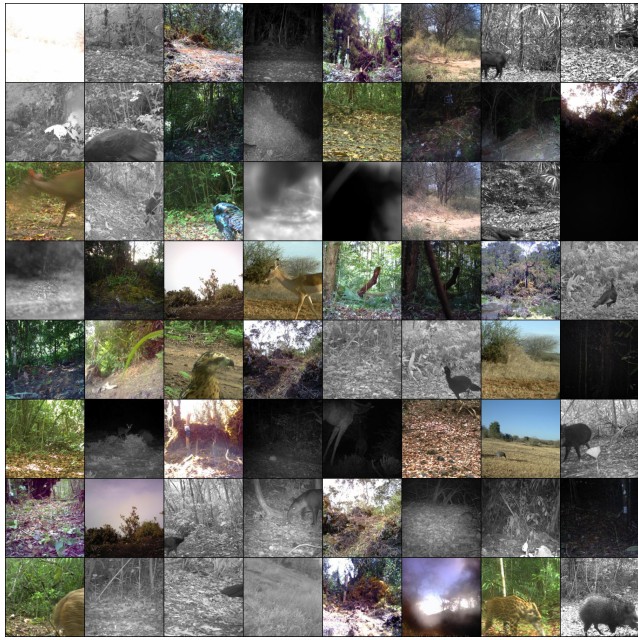

Figure 14: Random examples from the iWildCam-WILDS train set [22].

**ImageNet**    [37] We use ImageNet-1k from the ILSVRC 2012 challenge. It contains 1,000 diverse categories of animals and objects, with ∼1.2 million training images. The train set is roughly class balanced with ∼1.2 thousand images per category. The validation set contains 50,000 images and is exactly class balanced, with 50 images per class. Figure 15 shows examples of train set images.

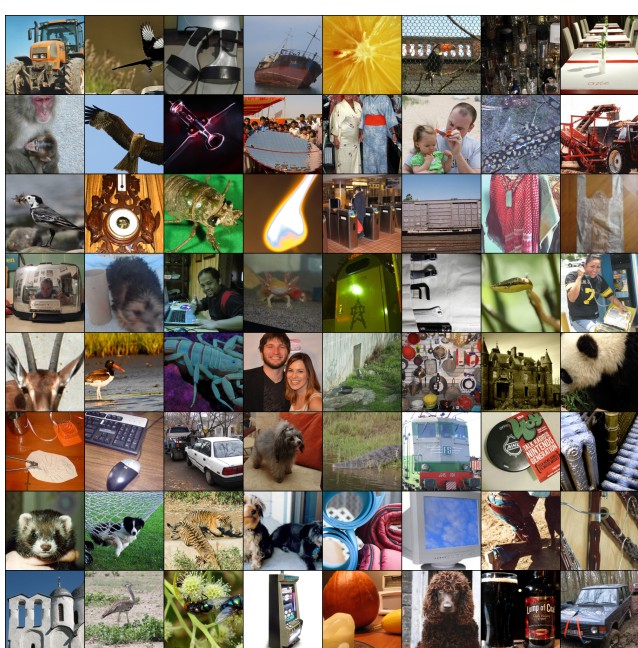

Figure 15: Random examples from the ImageNet ILSVRC 2012 challenge train set [37, 11].

**iNaturalist**    [46] We use the version of iNaturalist from the 2017 challenge, with 579,194 training images across 5,089 diverse natural organism categories. The full training set is notably not class balanced, exhibiting a long-tailed distribution (see Figure 16). The validation set contains 95,986

images and is also not class balanced, with between 4 and 44 images per category. Refer to Figure 17 for examples of iNaturalist data.

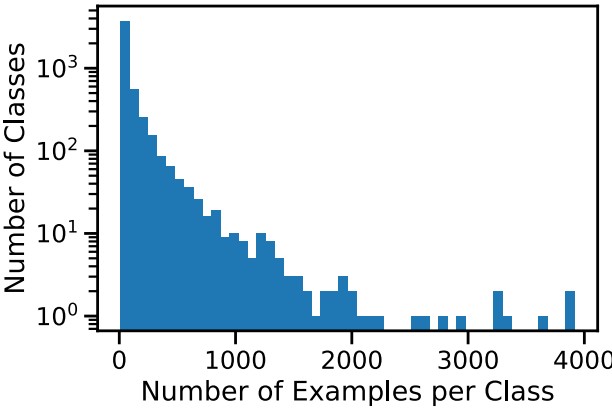

Figure 16: Histogram of class size distribution for the iNaturalist dataset.

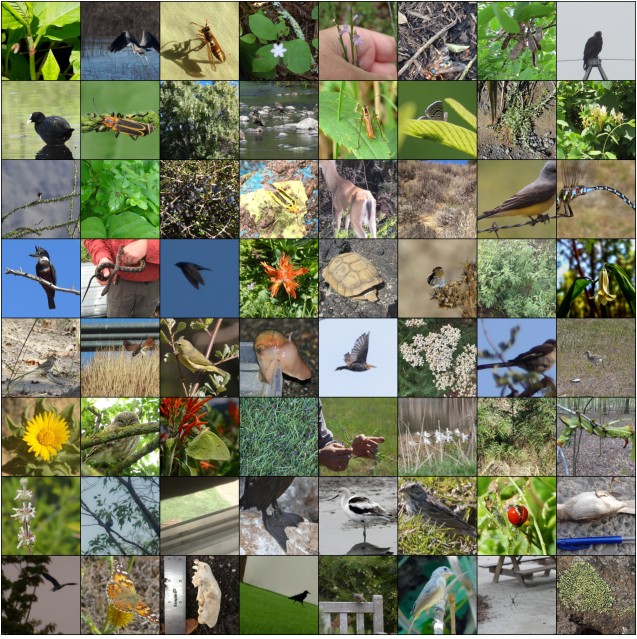

Figure 17: Random examples from the iNaturalist 2017 challenge train set [46].

# C  Additional Experiment Results

## C.1  Image Diversity

**More diverse data or more data per class?**    As described earlier in Section 4.4.1, we vary image diversity by randomly selecting pre-training images from a subset of classes in ImageNet, while keeping the total data budget fixed at 300K. Similar to the findings reported in Section 4.4.1, in Figure 18, we find that varying the number of classes used does not have an effect on the effective robustness of fine-tuned models.

| Superclass Category | Counts |
|---|---|
| Protozoa | 308 |
| Chromista | 398 |
| Actinopterygii | 1982 |
| Arachnida | 4873 |
| Animalia | 5228 |
| Fungi | 5826 |
| Mollusca | 7536 |
| Amphibia | 15318 |
| Mammalia | 29333 |
| Reptilia | 35201 |
| Insecta | 100479 |
| Plantae | 158407 |
| Aves | 214295 |

Table 2: Superclass data counts for the iNaturalist [46] train set. We use the 7 largest classes for our iNaturalist experiment in Section 4.4.2 so that we could select a uniform number of images per superclass while still having 80K images in total, to match the corresponding ImageNet experiment. Note that some superclass categories are in fact semantic superclasses of other categories (e.g. Aves, Reptilia, and Mammalia are all subclasses of Animalia) but are labeled disjointly. Given that we only use the last 7 categories, our superclass selection avoids this problem.

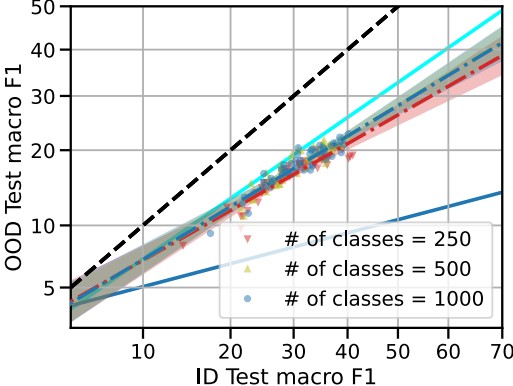

Figure 18: We repeat the experiment in Figure 7 with a higher data regime. With a total data budget of 300K images, we vary the number of classes randomly selected from the original 1000 ImageNet classes and adjust the number of images sampled from each class correspondingly. We observe that having 4× more classes, or 4× more images per class, induces the same level of robustness in fine-tuned models.

**Per-class diversity.** For experiments with the BREEDS [39] subset of ImageNet, we select 16 superclasses, each having 12 subclasses. The mapping can be found in Table 3. We choose overlapping subsets when modifying our diversity ratio (e.g., the 4 subclasses per superclass chosen when $p = 0.33$ are also part of the 8 subclasses per superclass chosen when $p = 0.66$).

For iNaturalist experiments, as mentioned in Section 4.4.2, we choose the 7 largest classes so that we could select a uniform number of images per superclass while keeping the total data budget at 80K, matching the setup for ImageNet. When modulating our diversity ratio, we randomly select a subset of subclasses from each superclass and uniformly sample images *with replacement* from these subclasses. As in the ImageNet experiment, subsets of subclasses chosen per superclass are overlapping at increasing diversity ratios.

| Superclass | Subclasses |
|---|---|
| garment | trench coat, abaya, gown, poncho, military uniform, jersey, cloak, bikini, miniskirt, swimming trunks, lab coat, brassiere |
| bird | African grey, bee eater, coucal, American coot, indigo bunting, king penguin, spoonbill, limpkin, quail, kite, prairie chicken, red-breasted merganser |
| reptile, reptilian | Gila monster, agama, triceratops, African chameleon, thunder snake, Indian cobra, green snake, mud turtle, water snake, loggerhead, sidewinder, leatherback turtle |
| arthropod | rock crab, black and gold garden spider, tiger beetle, black widow, barn spider, leafhopper, ground beetle, fiddler crab, bee, walking stick, cabbage butterfly, admiral |
| mammal, mammalian | Siamese cat, ibex, tiger, hippopotamus, Norwegian elkhound, dugong, colobus, Samoyed, Persian cat, Irish wolfhound, English setter, llama |
| fish | sturgeon, stingray, coho, great white shark, lionfish, gar, goldfish, hammerhead, rock beauty, anemone fish, barracouta, electric ray |
| accessory, accoutrement, accouterment | shower cap, stole, wig, Windsor tie, bib, necklace, bow tie, gasmask, umbrella, cowboy boot, Christmas stocking, bathing cap |
| appliance | espresso maker, sewing machine, stove, waffle iron, rotisserie, lawn mower, electric fan, dishwasher, iron, microwave, vacuum, space heater |
| craft | gondola, schooner, liner, airship, speedboat, airliner, canoe, trimaran, balloon, lifeboat, yawl, submarine |
| equipment | photocopier, mouse, dumbbell, iPod, home theater, projector, cassette player, hand-held computer, tennis ball, tape player, snorkel, monitor |
| furniture, piece of furniture, article of furniture | bookcase, throne, barber chair, four-poster, desk, medicine chest, crib, chest, sliding door, toilet seat, rocking chair, wardrobe |
| instrument | chime, wine bottle, ladle, reel, wall clock, hammer, wok, abacus, assault rifle, projectile, safety pin, corkscrew |
| man-made structure, construction | lumbermill, scoreboard, monastery, church, tobacco shop, drilling platform, dam, fountain, bell cote, yurt, bookshop, prison |
| wheeled vehicle | convertible, oxcart, electric locomotive, tricycle, fire engine, bicycle-built-for-two, moving van, golfcart, steam locomotive, jinrikisha, tractor, ambulance |
| cooked food, prepared food | consomme, burrito, meat loaf, bagel, ice cream, French loaf, ice lolly, hot pot, cheeseburger, potpie, trifle, mashed potato |
| produce, green goods, green groceries, garden truck | broccoli, cauliflower, mushroom, artichoke, acorn, head cabbage, spaghetti squash, jackfruit, cucumber, orange, hip, banana |

Table 3: Subclass-superclass mapping for the subset of ImageNet that we use to control per-class diversity, constructed based on the BREEDS hierarchy [39].

## C.2 CLIP Fine-tuning Details

We investigate how the effective robustness obtained from ImageNet pre-training would change at a much larger pre-training data regime. To do so, we fine-tune CLIP [32] models that have been trained on different data distributions on iWildCam-WILDS. We use the same finetuning hyperparameters from [22] with the AdamW optimizer [26]. The training datasets for CLIP are often large corpora of image-text pairs scraped from the internet. Models from the original CLIP paper [32] are trained with 400M data points, and we analyze specifically those that use ViT [12] as the image encoder architecture. We also include the results from CLIP ResNet50 models trained on YFCC-15M [45] and LAION-15M [40] separately, to show the effects of pre-training with different data scales and data sources. The results of these experiments can be found in Figure 12. We find that the linear trend of ImageNet pre-trained models is still predictive of the effective robustness obtained from these CLIP models. Note that along with utilizing significantly more data, CLIP also undergoes a different pre-training mechanism (i.e., contrastive versus supervised learning). This demonstrates the widespread applicability of existing effective robustness trends in analyzing generalization properties of pre-trained models.

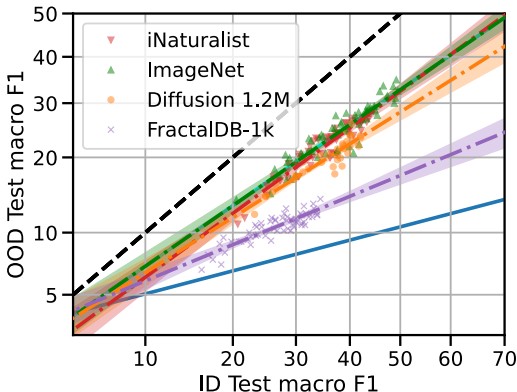

Figure 19: We experiment with pre-training on an ImageNet-scale synthetic dataset (i.e., 1.2M images) generated using Stable Diffusion, and find that it yields less robustness after fine-tuning compared to pre-training on natural data of similar quantities (i.e., ImageNet, iNaturalist). However, the natural-looking synthetic data in Diffusion 1.2M is still much more effective as a pre-training source compared to the synthetic fractal data in FractalDB-1k.

### C.3  Pre-training on ImageNet-scale Diffusion Data

As described in Section 4.5, we explore the effect of pre-training on images from different data distributions, both synthetic and natural. At 150K data regime (Figure 11), our earlier findings demonstrate that the transfer robustness obtained from pre-training on natural-looking synthetic data could rival that of pre-training on ImageNet or iNaturalist. Now, we increase the number of synthetic images generated using Stable Diffusion to 1.2M, matching the full ImageNet class distribution. At this data regime, we find that natural-looking synthetic data (i.e., Diffusion 1.2M) is slightly less effective than natural data (i.e., iNaturalist and ImageNet) at improving the effective robustness of fine-tuned models (Figure 19). However, Diffusion 1.2M still offers much more robustness gain compared to pre-training on synthetic fractal data (i.e., FractalDB-1k).

We hypothesize that the additional image diversity obtained from using more synthetic data saturates after certain data quantity, as the objects generated by Stable Diffusion appear to be relatively more centered and less cluttered, compared to ImageNet data (Figure 9). Further study into alternative prompting templates/ techniques is needed to increase the variety of the images obtained from Stable Diffusion.

## D  Investigating the effects of model size on robustness

We also examine how model size affects robustness, an important potential confounding variable which could change our analysis. One might suspect that larger models need more data to gain the same degree of robustness as smaller models. To do so, for the architectures that we experiment with, we plot model parameter sizes by their differences from effective robustness trends (points). If it was the case that models with more parameters needed more data to be robust, then we would expect the average of these residuals (lines) to be downward sloping. However, in Figure 20, we observe no such trend, and the average residuals are close to 0, demonstrating that model size has a negligible effect on robustness trends.

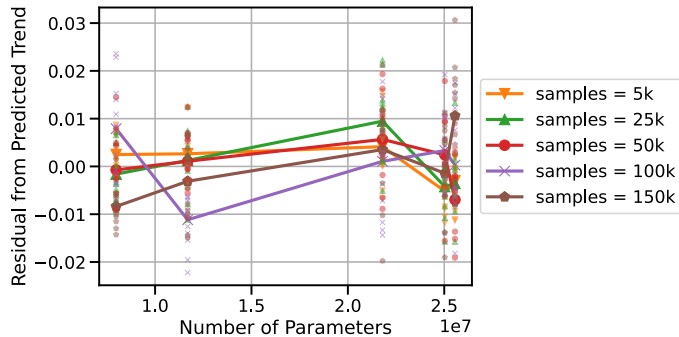

Figure 20: **Investigating the bias of model size on robustness.** We show the relationship between the parameter count of each architecture studied in Figure 4 (left) of our paper and their deviation from the overall effective robustness trend-line. We find that across both parameter count and amount of pre-training data, the average residual (line) remains consistent.

# E   Experiments with Contrastive Pre-training

We attempt to apply our interventions to the constrastive learning regime, done on image-text pairs scraped from the web. In this set of experiments, we construct different pre-training distributions from LAION-5B [40], and evaluate the resulting models *zero-shot* on ImageNet and two ImageNet-derived distribution shifts: ImageNet-V2 [33] and ImageNet-Sketch [47]. We experiment with pre-training CLIP ViT-B/32 on 10M, 50M and 100M samples separately, for 128M steps in total with batch size 4096 per GPU, learning rate 0.0005 and 500 warmup steps. In Figure 21, varying pre-training set size by 10× from 10M to 100M doesn't significantly alter the robustness linear trends on either distribution shift, despite offering substantial boost in robustness compared to training models from scratch on ImageNet (yellow line, obtained from [27]). We hypothesize that much larger scale differences are needed to test the effect of data quantity in this case. This additional investigation is not the focus of this work, however we believe adapting our interventions to massive pre-training webdatasets to be an interesting direction for future exploration.

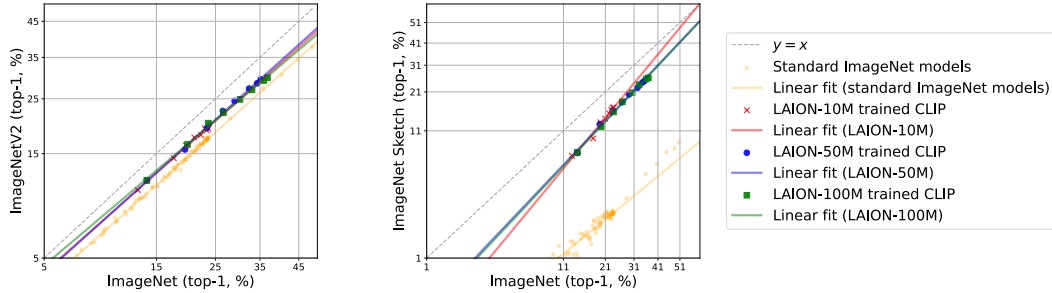

Figure 21: We experiment with pre-training with different data quantities sampled from LAION-5B [40], and find that 10M, 50M and 100M dataset sizes do not change downstream robustness when evaluated zero-shot on ImageNet-derived distribution shifts. However, pre-training still yields much more robustness compared to training from scratch (linear trend obtained from [27]), especially on the ImageNet to ImageNet-Sketch shift.

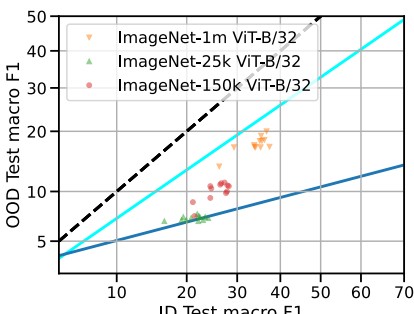 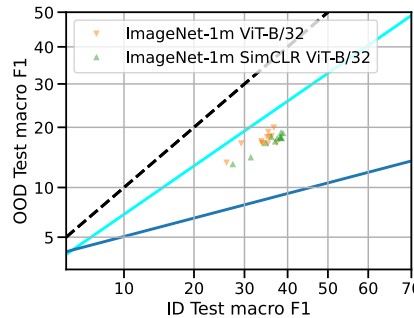

Figure 22: **ViT and self-supervised experiments. (left)** We show results of pre-training a ViT-B/32 on various quantities of ImageNet and **(right)** We compare ViT-B/32 pre-trained in a supervised manner on the full ImageNet dataset compared with a ViT-B/32 pre-trained with SimCLR on all of ImageNet. We do not show linear trends in this case because we only study one architecture here. We observe that vision transformers generally have lower effective robustness compared to smaller convolutional architectures for the same data quantity, but overall the finding of less pre-training data leads to lower effective robustness still holds. We further see that SimCLR pre-training exhibits similar robustness properties to supervised pre-training with the same data quantity (1M).

## F   Vision Transformers and Self-supervised Learning

In Figure 22 (left), we reproduce the data quantity analysis previously performed on CNNs with ViTs [12], an architecture which has become more widely used in recent years. Using the same models, in Figure 22 (right), we include analysis comparing supervised pre-training with that of SimCLR, a popular self-supervised training method [6]. Overall we observe that while ViTs generally demonstrate less robustness in this data regime than CNNs, our previous findings still hold—namely, larger data quantity improves fine-tuning robustness. This also applies to the SimCLR setting.

## G   DomainNet Experiments

For each domain provided in DomainNet [30], we compare models trained from scratch on data from that domain, to models that have been pre-trained on ImageNet and then fine-tuned on the same domain. We then evaluate the model performance on all other domains that have not been used for training, which are considered OOD test sets. Figure 23 shows the resulting effective robustness for all pairs of domains, with $x$ axis and $y$ axis representing ID and OOD accuracies respectively. We find that for most pairs, ImageNet pre-training and training from scratch do not exhibit distinctive linear trends. An exception to this can be found in the case of fine-tuning on Infograph data and evaluating on Sketch data (middle panel of the last row). Using this setup, we proceed to evaluating the impact of intervening on two different aspects of the ImageNet pre-training distribution that have been found to be important in earlier experiments with iWildCam-WILDS: label granularity and data quantity.

### G.1   Label Granularity

Our setup follows a similar procedure as described previously in Section 4.2. In the left plot of Figure 24, when we collapse the label space to 232 superclasses (depth 7), or 17 superclasses (depth 4), the effective robustness obtained from ImageNet pre-training is reduced to about the same level as training from scratch. This suggests that the Infograph-Sketch distribution shift is much more sensitive to label granularity, compared to the iWildCam-WILDS shift.

### G.2   Data Quantity

We repeat the experiment with pre-training data quantity described in Section 4.1. As seen from the right plot of Figure 24, using 25K, 50K, 100K and 150K subsets of ImageNet for pre-training results in a similar level of effective robustness after fine-tuning, which is a lot less than using the full ImageNet dataset, but still an improvement compared to training from scratch. As a sanity check, we

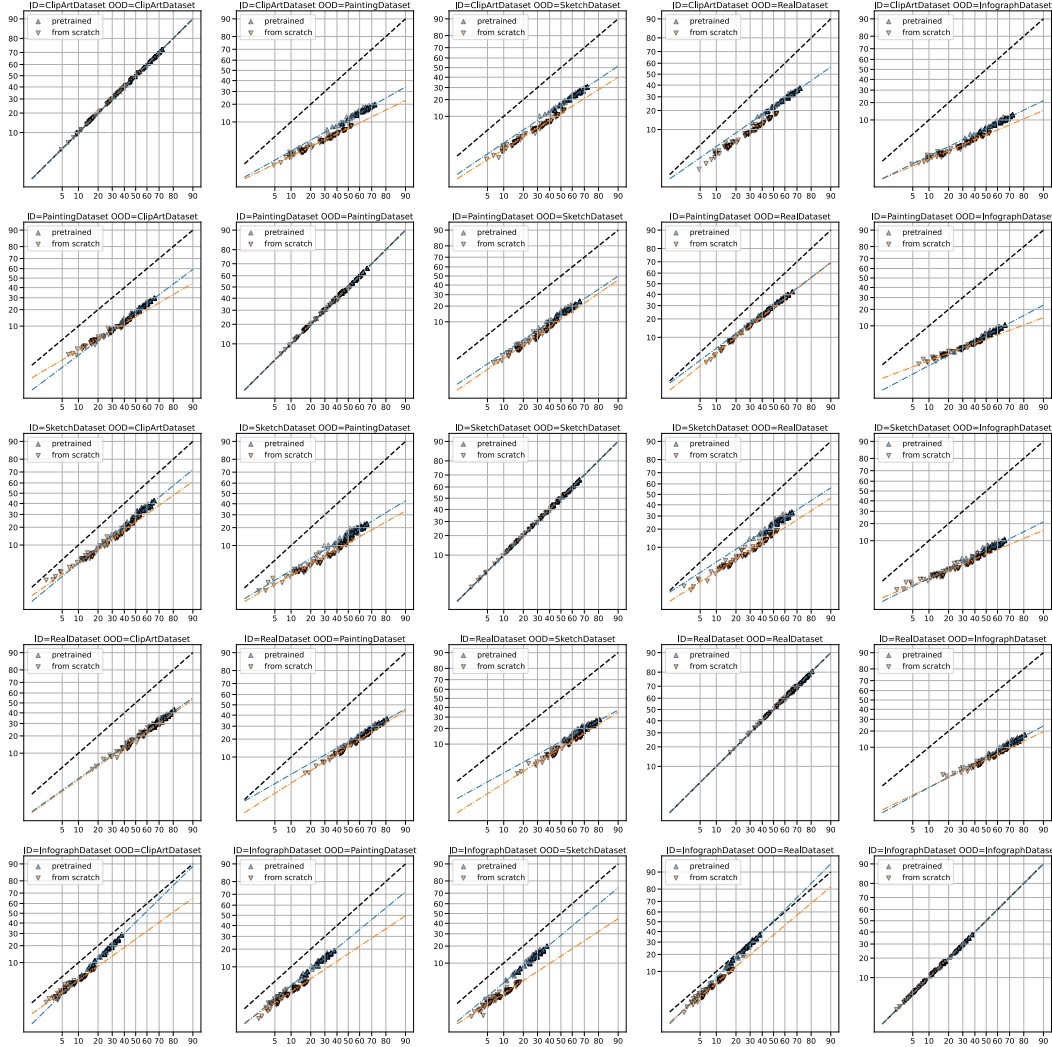

Figure 23: We compare effective robustness of models pre-trained on ImageNet and those trained from scratch on each domain (see each row), using the data from the same domain as ID test set, and data from every other domain as the OOD test set (see each column). In most cases, ImageNet pre-training and training from scratch yield similar downstream robustness, except for the Infograph & Sketch pair where we observe a sufficiently large gap in effective robustness.

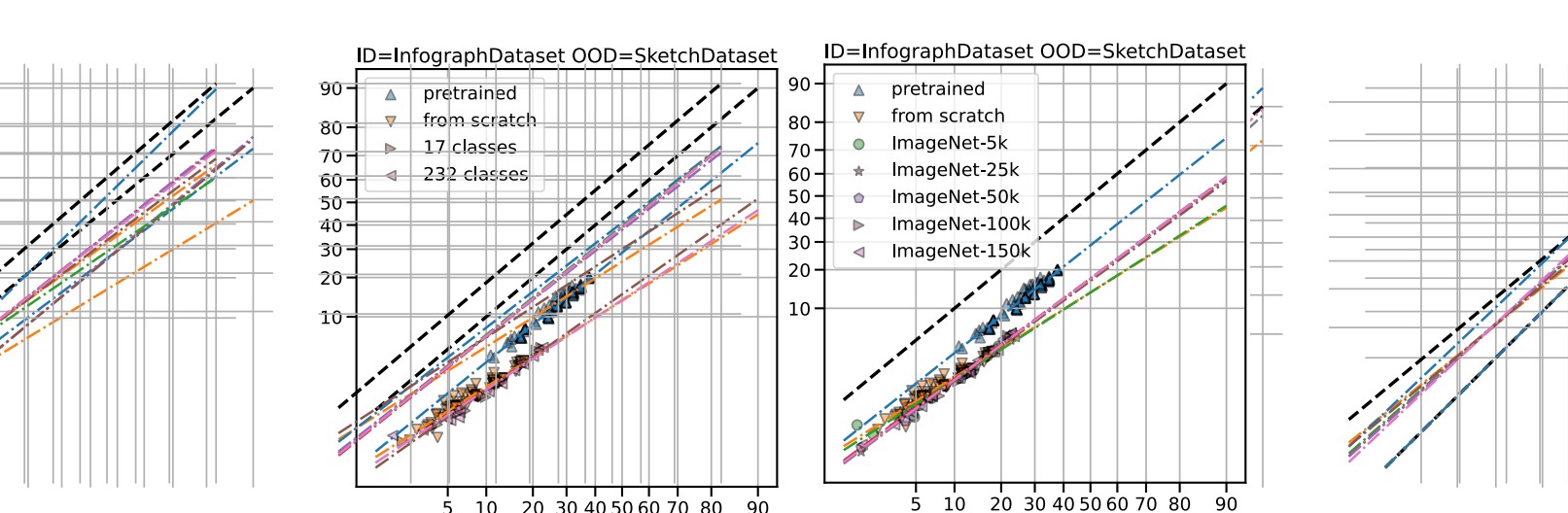

Figure 24: **(left)** Effect of label granularity. **(right)** Effect of data quantity. We perform the same interventions described earlier in Sections 4.1 and 4.2, on these two aspects of the ImageNet pre-training distribution that have been shown to be influential to downstream robustness. The ID and OOD data comes from Infograph and Sketch domains respectively. We again find that using more coarse labels and smaller data quantity during pre-training lowers the effective robustness of fine-tuned models significantly.

find that using only 5K ImageNet images for pre-training leads to no robustness gain compared to training from scratch on Infograph dataset itself.