# OpenReview forum: "On the Connection between Pre-training Data Diversity and Fine-tuning Robustness"
_NeurIPS.cc/2023/Conference — NeurIPS 2023 spotlight_

### Official Review · Reviewer_t8Yd · 2023-07-03

**Soundness:** 2 fair
**Presentation:** 2 fair
**Contribution:** 3 good
**Rating:** 6
**Confidence:** 3

**Summary:**

The paper introduces an empirical study for visual pre-training. By focusing on the pre-training, the authors conduct wide-range experiments through (i), data quantity, (ii) label granularity, (iii) label semantics, (iv) image diversity, (v) data sources. The empirical study considers how to use pre-training datasets (both real and synthetic datasets), fine-tuning (data distribution), and test tasks (image classification, out-of-distribution detection). In the experimental section, the study disappears several findings for visual pre-training.

**Strengths:**

- The novelty of the project is experimental findings with a wide-range of visual pre-training. The paper reveals five items in effect of data quantity (Section 4.1), label granularity (Section 4.2), label semantics (Section 4.3), image diversity (Section 4.4), and pre-training with different data sources (Section 4.5).

**Weaknesses:**

- How about implementing vision transformer (ViT) architecture? According to the experimental setup in Section 3, almost all of the network architectures are based on CNN architecture. Although some additional results are reported CLIP, one of the current visual architectures is undoubtedly ViT architecture. Since many architectures are implemented in the experiments, it doesn't seem that there is a lack of computing resources for the experimental conduction.

- The reviewer would touch the results shown in Figure 4. Does the graph show the highest score in the upper right corner? The claimed point with 25k pre-training samples (green line) is better than the baseline with 129k iWildCam images, but still wouldn't it be better to have more data such as 100k and 150k in terms of robustness? With this results, can we conclude tha "we do not need a lot of pre-training data to see significant robustness gains" (l.46-47)?

- Related to Section 4.5.1 and Figure 10, the PixMix framework [Hendrycks+, CVPR22] with the combination of real and synthetic images improves a robustness score. Therefore, for robustness performance, the present paper should implement PixMix framework rather than that of single usage of fractal images.

[Hendrycks+, CVPR22] Dan Hendrycks et al. "PixMix: Dreamlike Pictures Comprehensively Improve Safety Measures" in CVPR, 2022.

**Questions:**

- See above-mentioned questions

**Limitations:**

There are no negative limitations and societal impacts.

---

> ### Author Rebuttal · Authors · 2023-08-10
>
> We thank the reviewer for their time and feedback.
>
> **“How about implementing vision transformer (ViT) architecture?”** We understand that ViTs have gained popularity in recent years. We include results with ViTs in Figure 2 of our rebuttal PDF, specifically ViT-B/32, pre-trained in both supervised and self-supervised (SimCLR) manners. We show that our data quantity result holds for the ViT-B/32 architecture, and that supervised and self-supervised pre-training yields the same linear trends. Due to time constraints, we could not incorporate other ViT architectures for our current suite of experiments, however we will include the ViT-B/32, B/16, and L/14 architectures for subsequent versions of the paper.
>
> **“Does the graph show the highest score in the upper right corner?... With this results, can we conclude that ‘we do not need a lot of pre-training data to see significant robustness gains’ (l.46-47)?”** To clarify your question, the highest score is in the upper right hand corner, and we plot in-distribution and out-of-distribution performance of the same model. Perfect robustness is having the same performance on both distributions (the dotted black line in our plots). However note that a low-accuracy random classifier would also have perfect robustness, so there’s some nuance to this statement.
>
> The statement in Lines 46-47 does not contradict our findings regarding the impact of pre-training data quantity (Section 4.1). Of course pre-training on significantly more data will lead to further robustness improvement on downstream tasks (e.g. 150K versus 25K trend lines, see Figure 4). However, in Lines 46-47, we want to highlight that even with access to limited pre-training data (e.g. 25K), it is still beneficial to do pre-training, rather than training from scratch on the target task.
>
> **“For robustness performance, the present paper should implement PixMix framework rather than that of single usage of fractal images.”** Thank you for the pointer to this paper. The goal of our experiments with data sources (Section 4.5) is to disentangle the impact of the type of images (e.g. synthetic vs real), as well as the varying characteristics of the data distributions (e.g. long-tailed vs class-balanced). We acknowledge that there exist strategies to better combine signals from both real and synthetic sources to improve robustness. However, exploring data mixing strategies is beyond the scope of our investigation.

---

> > ### Comment · Area_Chair_wyE5 · 2023-08-20
> > **To Reviewer t8Yd: Please respond to the author rebuttal**
> >
> > Dear Reviewer t8Yd,
> >
> > The deadline for author discussion period is approaching soon. Please respond to the author's rebuttal and indicate whether your concerns have been addressed. Thank you!
> >
> > -AC

---

> ### Comment · Reviewer_t8Yd · 2023-08-21
> **Response to Author Rebuttal**
>
> Thank you so much for the authors. The authors response has addressed my questions. I will lean to the accept-side paper rating. Thanks again for the author's efforts.

---

### Official Review · Reviewer_ev5Y · 2023-07-03

**Soundness:** 4 excellent
**Presentation:** 3 good
**Contribution:** 4 excellent
**Rating:** 6
**Confidence:** 4

**Summary:**

This paper delves into the impact of pre-training data construction on fine-tuning robustness, encompassing aspects such as dataset size, class granularity, in- and out-class diversity, class similarity, and the use of synthetic data for pre-training. The evaluation metric employed is the in- and out-of-distribution testing performance, assumed to follow a linear trend. The study investigates changes in the slope of this trend to assess the influence of the mentioned factors. The findings suggest that pre-training data quantity and label granularity significantly affect fine-tuning robustness.


**Strengths:**

This paper presents a comprehensive exploration of the relationship between pre-training datasets and fine-tuning robustness. The novel perspective on dataset construction offers valuable insights into deep learning models.

**Weaknesses:**

A potential limitation lies in the reliance on a single metric to measure effective robustness, which might not provide a fully comprehensive evaluation of the connection between pre-training datasets and fine-tuning robustness.

The figures' lines and points lack sufficient definition, which makes it somewhat challenging to grasp their full meaning. Some lines appear to be extrapolated, raising concerns about their validity beyond the tested environments, as observed in Figure 4, 5, 6, 7, 8, 10, and 11.

**Questions:**

In addition to the general limitations mentioned in the ''Paper Weakness'' part, I have several additional questions:

1. Incorporating transformer networks into the experiments would be beneficial, considering their increasing popularity in vision tasks.

2. Addressing the influence of networks in the experiments, particularly in Sec. 4.1 "Effect of Data Quantity," is crucial. Larger models might suffer from reduced training dataset sizes, while smaller models could perform well under such circumstances. It is essential to mitigate this influence, as the plotted performance directly relates to different models.

3. The absence of a discussion on the influence of long-tailed data distribution is noteworthy. Understanding whether imbalanced pre-training datasets affect fine-tuning robustness is crucial. Treating data distribution merely as a confounding effect might not be sufficient.

4. In some cases, such as Figure 4 and 11, the linear trend is not readily apparent, and the scattered data points seem to follow more irregular patterns. Further clarification on these trends and their implications would be helpful.

**Limitations:**

The authors have addressed some limitations of their work, and there are additional suggestions for improvement in the 'Paper Weakness'' part and "Questions" part.

---

> ### Author Rebuttal · Authors · 2023-08-10
>
> Thanks to the reviewer for providing valuable suggestions on how we can improve our work.
>
> **“The reliance on a single metric to measure effective robustness, which might not provide a fully comprehensive evaluation of the connection between pre-training datasets and fine-tuning robustness.”** Effective robustness trends have been shown to hold for a variety of interventions, including training method, hyperparameters and dataset size [2]. Besides, robustness could be defined in many different ways in existing literature. Given the various interventions we could perform on the pre-training distribution, we chose to focus on robustness to natural distribution shifts, which has become increasingly important in the context of large-scale pre-training.
>
> **“The figures' lines and points lack sufficient definition, which makes it somewhat challenging to grasp their full meaning.”** Thank you for the useful feedback. We provide background on the effective robustness framework as well as what the points represent in Section 2 (Background). We will improve the clarity as well as the context with respect to related work in subsequent revisions. The reviewer could also refer to previous work [1, 2] for more details.
>
> **“Some lines appear to be extrapolated, raising concerns about their validity beyond the tested environments, as observed in Figure 4, 5, 6, 7, 8, 10, and 11.”** Thank you for the important observation. Concerns around extrapolation are the reasons why we provide error bars for the reported linear trends in the form of bootstrap confidence intervals. Besides, across the range of performance currently attainable on the downstream task (iWildCam-WILDS), we find that the F1 scores resulting from our interventions overlap significantly, allowing us to fairly compare the robustness properties across different pre-training distributions that yield similar performance range.
>
> **“Incorporating transformer networks into the experiments would be beneficial, considering their increasing popularity in vision tasks.”** Per your comment, we report new results with ViTs in Figure 2 of our rebuttal PDF, specifically ViT-B/32, pre-trained in both supervised and self-supervised (SimCLR) manners. We observe that our data quantity result holds for the ViT-B/32 architecture. We will expand our analysis of ViT with more architectures in the subsequent versions of the paper.
>
> **“Larger models might suffer from reduced training dataset sizes, while smaller models could perform well under such circumstances. It is essential to mitigate this influence, as the plotted performance directly relates to different models.”** This is an interesting point to study. In Figure 1 (right) of the rebuttal PDF, we plot the residuals to the overall trend line of pre-training on a fixed data quantity (shown in Figure 4 of the main paper), for each architecture size. We find that the residuals do not exhibit any particular pattern and remain relatively consistent across different model sizes.
>
> **“The absence of a discussion on the influence of long-tailed data distribution is noteworthy. Understanding whether imbalanced pre-training datasets affect fine-tuning robustness is crucial.”**  We do include a discussion of pre-training on long-tailed data in our paper. Our experiments involve pre-training on two versions of iNaturalist: the raw, long-tailed data (Figure 10) and the 1000-class, class-balanced subset of iNaturalist using its most frequent classes (Figure 11). In both cases, we find that iNaturalist behaves similarly to ImageNet in terms of the robustness properties. We discuss these findings in Lines 239 - 248 of our paper.
>
> **“In some cases, such as Figure 4 and 11, the linear trend is not readily apparent, and the scattered data points seem to follow more irregular patterns.”** For certain experiments (e.g. pre-training with 50K or 100K images, Figure 4), our interventions make little difference on the downstream robustness. Consequently, the linear trends end up overlapping. Per the reviewer’s concern, we report the linear fit of each trend line in Figure 4 and Figure 11 in the following table. We note that the $R^2$ is high for all of the linear trends. The “irregular data pattern” effect is largely a result of the intervention not producing a large change in linear fit, making our data points seem like a large cluster. We will include separate plots for each subsample experiment as well as add this $R^2$ analysis to the appendix.
>
> | Pre-Training Dataset | Samples | Coefficient of Determination ($R^2$) |
> | ------------------- | ------- | ---------------------------------- |
> | Imagenet            | 5k      | 0.634                            |
> | ImageNet            | 25k     | 0.799                            |
> | ImageNet            | 50k     | 0.874                            |
> | ImageNet            | 100k    | 0.769                            |
> | ImageNet            | 150k    | 0.862                            |
> | iNaturalist         | 5k      | 0.689                            |
> | iNaturalist         | 25k     | 0.741                            |
> | iNaturalist         | 50k     | 0.873                            |
> | iNaturalist         | 100k    | 0.824                            |
> | iNaturalist         | 150k    | 0.908                            |
> | iNaturalist         | 150k    | 0.908                            |
> | ImageNet            | 150k    | 0.862                            |
> | Diffusion           | 150k    | 0.906                            |
> | FractalDB-1k        | 150k    | 0.565                            |
>
>
> [1] Measuring robustness to natural distribution shifts in image classification. Taori et al., 2020.
>
> [2] Accuracy on the Line: On the Strong Correlation Between Out-of-Distribution and In-Distribution Generalization. Miller et al., 2021.

---

> > ### Comment · Reviewer_ev5Y · 2023-08-18
> > **Response to Author Rebuttal**
> >
> > Thank you for your rebuttal; it has effectively addressed most of my concerns.

---

### Official Review · Reviewer_6SSm · 2023-07-04

**Soundness:** 3 good
**Presentation:** 3 good
**Contribution:** 3 good
**Rating:** 6
**Confidence:** 4

**Summary:**

In this paper, the authors investigate the influence  of pre-training data on the robustness of fine-tuning. The authors design several experiments by following a common pre-training, fine-tuning and evaluation pipeline. They  found that the quantity of the pre-training
data and the granularity of the label set  are two most influential factors on the robustness of downstream fine-tuning. The  authors further leverage synthetic data from Stable Diffusion to increase the effectiveness of the pre-training distribution along these two ablation axes.

**Strengths:**

1. The paper is  well-written and the motivation is clear.
2. The authors considers many factors that will possibly  affect the fine-tuning robustness including data quantity, label granularity, label  semantic, etc.
3. The conclusions of the paper are clear which is useful for follow-up research.

**Weaknesses:**

1. The  authors mainly focus on the impact of the  pre-training data on the fine-tuning robustness, it is not clear if different fine-tuning methods (only fine-tune the last layer, etc) would change the conclusions of the paper.
2. The authors considers out-of-distribution generalization in this paper, it is unclear whether the conclusions also apply to other concepts of robustness, such as adversarial robustness.

**Questions:**

1. In figure 3, why the fine-tuning epoch is 12?

2. Would the number of fine-tuning epochs affect the  robustness of the pre-trained model? The authors found that by only using 5K samples for pre-training offers no robustness gain, this raises the question that if longer fine-tuning will also reduce robustness.

**Limitations:**

Yes.

---

> ### Author Rebuttal · Authors · 2023-08-10
>
> We thank the reviewer for the constructive feedback!
>
> **“It is not clear if different fine-tuning methods (only fine-tune the last layer, etc) would change the conclusions of the paper.”** This would be a useful direction for future investigation. Our paper includes experiments with linear probes for CLIPs pre-trained on different data sources (Figure 12). There we find that the fine-tuned models lie on the same linear trend as models pre-trained on ImageNet and fine-tuned end-to-end. We will include experiments with different fine-tuning methods in the subsequent version of our paper. We expect this to follow similar linear trends as these kinds of training interventions have been studied previously in [1].
>
> **“The authors consider out-of-distribution generalization in this paper, it is unclear whether the conclusions also apply to other concepts of robustness, such as adversarial robustness.”** This is an interesting point. We are primarily interested in the study of natural distribution shifts as current paradigms for pre-training seek to have a very general backbone that encompasses a wide range of data, in order to be robust to many downstream natural distribution shifts (e.g. CLIP). We believe that having a better understanding of how to choose pre-training distributions to be robust to these kinds of shifts would be highly relevant. We agree that studying other robustness concepts is an interesting avenue for future work, but is not a focus of our current work, especially since adversarial robustness itself can be defined in many different ways in existing literature.
>
> **“In figure 3, why the fine-tuning epoch is 12?”** This choice follows from the experiment setup in previous work [1]. In Figure 3, using linear trends obtained from [1], we also study the impact of the number of fine-tuning epochs on how well the resulting model’s performance fits the linear trend, and find that varying the number of epochs does not change the linear trend significantly. We also include new experiments with longer fine-tuning (see Figure 1 (left) of the rebuttal PDF).
>
> **“The authors found that by only using 5K samples for pre-training offers no robustness gain, this raises the question that if longer fine-tuning will also reduce robustness.”** Per your comment, we have added new experiments fine-tuning models pre-trained on different data quantities (5K and 25K) for double the number of epochs. We observe that longer fine-tuning does not change the resulting trend lines. Refer to Figure 1 (left) of our rebuttal PDF for more details.
>
> [1] Accuracy on the Line: On the Strong Correlation Between Out-of-Distribution and In-Distribution Generalization. Miller et al., 2021.

---

> > ### Comment · Area_Chair_wyE5 · 2023-08-20
> > **To Reviewer 6SSm: Please respond to the author rebuttal**
> >
> > Dear Reviewer 6SSm,
> >
> > The deadline for author discussion period is approaching soon. Please respond to the author's rebuttal and indicate whether your concerns have been addressed. Thank you!
> >
> > -AC

---

### Official Review · Reviewer_XBK9 · 2023-07-06

**Soundness:** 3 good
**Presentation:** 3 good
**Contribution:** 3 good
**Rating:** 6
**Confidence:** 3

**Summary:**

This paper investigates the role of pre-training data diversity on fine tuning robustness. They vary various factors like label space, label semantics, image diversity, data domains, and data quantity of the pre-training distribution to investigate how these factors impact the robustness of the models. Some interesting insights include similar label semantics doesn't necessarily improve the robustness of the model and increasing per-class examples doesn't necessarily improve the robustness of the model.

**Strengths:**

- Provides insights on pre-training models like how various factors like data diversity, label space, label semantics etc affect the robustness of the models. Might be helpful for ML practitioners trying to decide what kind of data is best suited for pre-training
- Experiments are thorough and clean. Paper is also easy to read highlighting main results.



**Weaknesses:**

- Major section of the work has been focused on supervised pre-training which is becoming less and less common with the advent of self-supervised learning methods. It would have been interesting to look at these of pre-training strategies in much more depth.
- Only one downstream task considered in the experiments (iWildCam-WILDS). Hard to quantify if these results generalize to other datasets.

**Questions:**

- Can these results generalize to other downstream task as well? In this work only iWildCam-WILDS dataset is incorporated into results, it would be interesting to see if these results generalize to other datasets and datasets from different domains.

**Limitations:**

addressed.

---

> ### Author Rebuttal · Authors · 2023-08-10
>
> We thank the reviewer for the effort they have put into reviewing our paper.
>
> **“It would have been interesting to look at self-supervised learning methods in much more depth.”** This is a good direction for future work. The focus of our current work is on supervised pre-training. Per the reviewer’s feedback, we have included experiments with SimCLR. In Figure 2 (right) of the rebuttal PDF, we include a comparison of a ViT-B/32 model pre-trained with SimCLR on ImageNet and the same architecture pre-trained on ImageNet in a supervised fashion. We see that models trained on the same dataset exhibit similar robustness trends despite the differing pre-training strategies. We note that certain properties of the pre-training distribution that we examined are harder to control/not applicable in the self-supervised learning regime (e.g. label granularity), which is why it is not the main focus of the paper.
>
> **“Only one downstream task considered in the experiments (iWildCam-WILDS). Do these results generalize to other datasets?”** This is a notable limitation which we acknowledge in the Conclusion section. We do study some of the properties of the pre-training distribution in the context of DomainNet (see Appendix E) and show that our main findings on the impact of label granularity and data quantity hold. Overall, characterizing distribution shifts where pre-training on non-web-scale datasets can provide a significant boost in effective robustness compared to training from scratch is an important direction for future work. The effect of varying pre-training distributions on downstream effective robustness has been studied at larger scales across a variety of distribution shifts (e.g. [1]), but at smaller scales the only shift that consistently sees substantial robustness benefits from small-scale pre-training is iWildCam-WILDS [2], making it a useful testbed.
>
> [1] Quality Not Quantity: On the Interaction between Dataset Design and Robustness of CLIP. Nguyen et al., 2022.
>
> [2] Accuracy on the Line: On the Strong Correlation Between Out-of-Distribution and In-Distribution Generalization. Miller et al., 2021.

---

> > ### Comment · Area_Chair_wyE5 · 2023-08-20
> > **To Reviewer XBK9: Please respond to the author rebuttal**
> >
> > Dear Reviewer XBK9,
> >
> > The deadline for author discussion period is approaching soon. Please respond to the author's rebuttal and indicate whether your concerns have been addressed. Thank you!
> >
> > -AC

---

> > ### Comment · Reviewer_XBK9 · 2023-08-20
> > **Response to rebuttal**
> >
> > Thanks for addressing all the concerns raised and adding experiments with SimCLR. I will like to keep my score.

---

### Author Rebuttal · Authors · 2023-08-10

We thank the reviewers for their thorough, insightful comments and have made revisions based on their feedback. We are glad they found the work well motivated, novel, and the contributions to be of value to the community. Here we include new results and plots to respond to some of the concerns by reviewers, see the attached PDF. We are happy to engage in further discussion and interested in any additional feedback the reviewers may have.

---

### Comment · Area_Chair_wyE5 · 2023-08-18
**To Reviewers: Please respond to the author rebuttals.**

Dear reviewers,

Thank you for serving as a reviewer for NeurIPS!

We are towards the end of the discussion stage with authors, but some of you haven't posted your response to the author rebuttals yet.
As the scores for this paper are diverse, please check the author rebuttals, reply to them and update your score (when necessary) ASAP. Thanks!

-AC

---

### Decision · Program_Chairs · 2023-09-21

**Decision:**

Accept (spotlight)

**Comment:**

This paper presents a comprehensive empirical study examining the impact of pre-training data diversity on fine-tuning robustness. The conducted experiments encompass a wide array of factors, including data quantity, label granularity, label semantics, image diversity, and data sources. The resulting findings offer valuable insights for practitioners seeking to optimize pre-training models. The reviewers also raised concerns on its clarity and insufficient downstream task evaluation.
Overall, the meta-reviewer deems the contribution of this paper to be substantial and recommends it for publication. The authors are encouraged to incorporate the feedback provided by the reviewers when preparing the camera-ready version.